SOFTWARE

# miRScore: A rapid and precise microRNA validation tool

**Allison Vanek**[1,2], **Sam Griffiths-Jones**[3], **Blake C. Meyers**[4], **Saima Shahid**[5], **Michael J. Axtell**[1,2*]

**1** Bioinformatics and Genomics Ph.D. Program, Huck Institutes of the Life Sciences, The Pennsylvania State University, University Park, Pennsylvania, United States of America, **2** Department of Biology, The Pennsylvania State University, University Park, Pennsylvania, United States of America, **3** School of Biological Sciences, Faculty of Medicine, Biology and Health, Michael Smith Building, The University of Manchester, Manchester, United Kingdom, **4** Department of Plant Sciences, University of California, Davis, California, United States of America, **5** Plants, Photosynthesis and Soil, School of Biosciences, The University of Sheffield, Western Bank, Sheffield, United Kingdom

* mja18@psu.edu

## Abstract

MicroRNAs (miRNAs) are small non-protein-coding RNAs that regulate gene expression in many eukaryotes. Next-generation sequencing of small RNAs (small RNA-seq) is central to the discovery and annotation of miRNAs. Newly annotated miRNAs and their longer precursors encoded by *MIRNA* loci are typically submitted to databases such as the miRBase microRNA registry following the publication of a peer-reviewed study. However, genome-wide scans using small RNA-seq data often yield high rates of false-positive *MIRNA* annotations, highlighting the need for more robust validation methods. miRScore was developed as an independent and efficient tool for evaluating new *MIRNA* annotations using sRNA-seq data. miRScore combines structural and expression-based analyses to provide rapid and reliable validation of new *MIRNA* annotations. By providing users with detailed metrics and visualization, miRScore enhances the ability to assess confidence in *MIRNA* annotations. miRScore has the potential to advance the overall quality of *MIRNA* annotations by improving accuracy of new submissions to miRNA databases and serving as a resource for re-evaluating existing annotations.

## Author summary

MicroRNAs (miRNAs) play a major role in gene regulation in most eukaryotic organisms. Genome-wide analysis of miRNAs and miRNA-encoding precursors (here, *MIRNA*s), can lead to numerous false positive annotations. Criteria for *MIRNA* annotation often use a combination of short RNA sequencing and predicted RNA secondary structural properties of precursors. However, implementation of these criteria varies. Here, we introduce a tool, miRScore, developed to

**Data availability statement:** The miRScore software is an open-source python project available under the permissive MIT license. Documentation, source code, and test examples are freely available on the github page at https://github.com/Aez35/miRScore. Newly generated small RNA-seq data are available from NCBI GEO under accession GSE282265.

**Funding:** This work was supported by the National Science Foundation (2130884 to MJA and BCM; 2450802 to BCM), the Biotechnology and Biological Sciences Research Council (BB/W018438/1 to SGJ), and a seed grant from The Huck Institutes of the Life Sciences at Penn State (un-numbered award to MJA). The funders had no role in study design, data collection and analysis, decision to publish, or preparation of the manuscript.

**Competing interests:** The authors have declared that no competing interests exist.

standardize *MIRNA* validation using accepted annotation criteria. miRScore is intended to improve the quality of *MIRNA* annotation. miRScore takes as input one or more putative miRNA sequences, one or more corresponding precursor RNAs, and short RNA sequencing data. miRScore quickly and accurately evaluates each candidate in the context of the provided data and determines whether each annotation meets all criteria. These results can be used to determine high-confidence miRNAs for cataloging and downstream analysis.

## Introduction

MicroRNAs (miRNAs) are a class of small, non-coding RNAs that regulate gene expression within eukaryotes. This regulation typically occurs when a miRNA, which is loaded into an RNA-induced silencing complex (RISC), imperfectly base pairs to a target messenger RNA (mRNA). The RISC then frequently acts as an endonuclease to cleave the mRNA or to otherwise inhibit its translation [1–4]. miRNA-directed regulation of mRNAs is crucial in various biological processes such as developmental timing [5–7], metabolism [8,9], and defensive pathways [10–13] in both plants and animals. Although miRNA biogenesis varies somewhat between animals and plants, the fundamental aspects of miRNA structure and function are conserved [14]. In both plants and animals, the precursors of miRNAs are generally transcribed by RNA polymerase II from an endogenous *MIRNA* gene. While many *MIRNA* primary transcripts are transcribed as independent genes from intergenic regions, some are processed from the introns of protein-coding mRNAs [4]. Transcription results in a long single-stranded RNA containing a hairpin, called the primary miRNA (pri-miRNA). The hairpin embedded within the primary transcript is then processed by sequential endonuclease activity (Drosha and Dicer in animals, or by a single Dicer-Like protein in plants) to release a miRNA duplex. The miRNA duplex is a double-stranded RNA, typically with a few mismatched and/or bulged nucleotides, which consists of the mature functional strand (miRNA) and passenger strand (miRNA*). The miRNA duplex is unwound, and a single-stranded mature miRNA is bound to an Argonaute protein to form the RISC. Most frequently a single strand from this duplex is incorporated into RISC and regulates mRNAs; in some cases both strands from the miRNA duplex become separately bound to different RISCs and have two distinct constellations of mRNA targets. For details of microRNA biogenesis, see [4,15,16].

Alignment of deep small RNA-sequencing (sRNA-seq) data to a reference genome is a common method for *MIRNA* annotation and quantification. Several tools such as ShortStack [17,18], miRador [19], miRDeep [20], and miRDeep-P2 [21], have been developed to annotate miRNAs and other small RNAs using sRNA-seq data. These tools typically work by aligning sRNA-seq data to a reference genome, followed by evaluation of potential miRNA-encoding loci (*MIRNA)*. One way candidate *MIRNAs* are identified is by the distinctive alignment pattern of the miRNA/miRNA* duplex reads to the hairpin precursor. miRNA and miRNA* reads from sRNA-seq align to

a single genomic strand, as their precursors are single-stranded transcripts. These reads align a short distance from each other, forming two distinct "stacks" of read coverage [17,22]. *MIRNA* primary transcripts are typically short-lived and hard to detect using sRNA-seq or regular mRNA-seq. Most sRNA-seq centered *MIRNA* identification tools thus annotate "hairpin" sequences that encompass the stem-loop region and some adjacent sequence of pre-determined length. The start and stop positions of these annotations do not necessarily correspond to the ends of the actual primary transcripts. The secondary structure of this putative hairpin precursor is then predicted. For true *MIRNAs*, the predicted secondary structure of the putative precursor RNA is an imperfect stem-loop. Furthermore, two stacks of aligned sRNA-seq reads from the miRNA and the miRNA* are found on opposite arms of the predicted stem-loop with a diagnostic two nucleotide 3'-overhang. Generally, the sequence with the most abundant set of reads is termed the 'mature' miRNA, while the sequence with less abundant reads is the 'star' sequence. Detection of reads from both arms of the miRNA duplex is required to confirm the predicted duplex [23–25]. Identification of candidate *MIRNAs* using sRNA-seq is therefore dependent on empirical evaluation of read alignment patterns in the context of the presumed precursor's predicted RNA secondary structure.

The identification of *MIRNAs* through deep sequencing data poses some challenges. One is the handling of multimapping reads, in which there are multiple best-scoring alignments for a single read. This occurs frequently with sRNA-seq data due to shorter read lengths and the fact that identical miRNAs can be encoded by paralogous loci [18]. Another challenge is distinguishing true *MIRNAs* from other sRNA classes such as short-interfering RNAs (siRNAs), which have their own unique alignment patterns and criteria [23,26]. Each *MIRNA* discovery tool employs distinct methods for handling these challenges, with varying degrees of performance for identification of novel *MIRNAs* in plants and animals [17,19–21]. The lack of uniform implementation of well-defined *MIRNA* criteria, coupled with the challenging nature of informatically distinguishing miRNAs from noise or other sRNA species, has led to diminishing confidence in the overall quality of existing *MIRNA* annotations [24,27–30].

There have been considerable efforts to define *MIRNA* criteria to improve the quality of annotations [23–25,31]. Some miRNA databases contain a significant number of false positive annotations [28,30,32]. miRBase for example relies on researchers and peer reviewers to assess the validity of miRNAs before submission, and has adopted methods of determining confidence in these community-based annotations [27,32]. The current release of miRBase (V. 22.1) contains over 48,000 mature miRNA sequences from 271 diverse species including animals, plants, and some protists [32]. MirGeneDB has taken a different approach, manually curating *MIRNA* annotations of metazoan species through structural, expression, and conservation analysis [25,31,33]. Whether by database curators or the research community, the assessment of novel miRNAs relies on a degree of manual inspection and evaluation. However, manual inspection of incoming annotations takes significant effort and currently lacks standardized implementation.

While there are many *de novo* miRNA annotation tools and miRNA databases available, a secondary method to quickly analyze novel and annotated *MIRNAs* following genome-wide sRNA annotation is not available. Such a tool, to rapidly check new annotations, could be useful for database curators by removing the need for time-consuming manual inspection of new submissions. A standardized and quick method of automatically validating new *MIRNA* annotations would improve the quality of annotations published and subsequently submitted to online repositories. Retrospective application of such a method could also be used to flag and remove problematic entries in repositories such as miRBase. To address this need, we developed miRScore – a rapid and precise miRNA validation tool. miRScore can rapidly evaluate the annotation of both existing and novel miRNAs against specific sRNA-seq datasets using widely accepted *MIRNA* criteria in plants and animals. It offers a comprehensive evaluation of *MIRNA* loci, analyzing each criterion and producing visualizations of hairpin secondary structure and expression patterns. In this study, miRScore is described and tested using both annotated and novel *MIRNAs* from plants and animals.

## Design and implementation

miRScore is implemented as a Python script that requires several commonly used bioinformatic tools including samtools [34], ViennRNA [35], and bowtie [36]. miRscore is an open-source software available under a permissive MIT license from GitHub at https://github.com/Aez35/miRScore, and is easily installed using Bioconda [37].

### Workflow

miRScore validates *MIRNA* loci by analyzing the hairpin precursor sequence, miRNA duplex, and sRNA-seq data. The validation process is based on a set of previously described criteria which can be categorized as either structural (based on the predicted RNA secondary structure of the precursor) or expression (based on observations of miRNA and miRNA* abundance) (Table 1) [23–25,27,38,39]. miRScore utilizes a "pass/fail" system of reporting: each input *MIRNA* locus will ultimately either pass or fail. In some cases, one or more warnings may be raised for a "passed" entry if certain atypical features are present. A full list of flags and their explanations can be found in Table 2 and in the miRScore README. If one or more flags with a "fail" result are present, the locus will fail. Loci with no flags will pass, as will loci that have one or more flags associated with a "warning" result but no flags with a "fail" result (Table 2).

The primary use of miRScore is to rapidly assess novel *MIRNA* annotations prior to publication and submission to miRNA databases (Fig 1A). Users input properly formatted FASTQ or FASTA files containing sRNA sequencing reads, as well as miRNA duplex sequences and hairpin sequences in FASTA format (Fig 1B). The precursor sequences should be extended past the endonuclease cut sites, and the miRNA/miRNA* should not start or end the precursor sequence. This is to allow proper evaluation of the miRNA duplex structure. The identifier of each hairpin should be nearly identical to the corresponding mature miRNA, with the exception being that the mature identifier may contain '3p', '5p', or 'mature' and still be discerned. Users may also include miRNA* sequences in the mature FASTA file, provided they be distinguishable from the mature sequence by the either a "-5p","-3p", ".star", or "*" at the end of the name (i.e. miR399-3p, miR399.star, miR399*) (Fig 1C). The workflow of miRScore is to evaluate structural and expression criteria of all loci, assign a pass or fail result, reanalyze each failed locus for potential rescue (see below), and generate visualizations.

### Structural evaluation

miRScore predicts the secondary structure of single-stranded hairpin precursors using RNAfold from ViennaRNA [35]. The location of the miRNA and miRNA* sequences are indexed on the hairpin. If the user does not provide a miRNA* sequence, miRScore predicts it by determining the sequence that forms a miRNA/miRNA* duplex with a two-nucleotide 3' overhang. miRScore then evaluates the miRNA duplex and hairpin against structural criteria (Table 1). This predicted secondary structure is used to determine characteristics such as the number of mismatches and large bulges within the duplex, or whether there is a two-nucleotide 3' overhang.

**Table 1. Criteria for endogenous miRNAs in plants and animals.**

| | Criteria | Category |
|---|---|---|
| 1 | One miRNA/miRNA* duplex with a two-nucleotide 3' overhang | Structural |
| 2 | Up to five mismatched (plants) or seven mismatched (animals) nucleotides in the miRNA/miRNA* duplex | Structural |
| 3 | No asymmetric bulge larger than three nucleotides | Structural |
| 4 | miRNA and miRNA* between 20–24 nucleotides for plants or 20–26 nucleotides for animals | Structural |
| 5 | At least ten reads must align perfectly to the miRNA/miRNA* | Expression |
| 6 | At least 75% of reads must align to the miRNA duplex and one-nucleotide positional variants (precision) | Expression |

**Table 2. List of potential miRScore flags and their consequences.**

| Flag | Explanation | Result if flag is present |
|---|---|---|
| More than 5 mismatches in duplex | More than 5 base pairs are mismatched in the miRNA duplex. | Fail (plants)/ Warning (animals) |
| More than 7 mismatches in duplex | More than 7 base pairs are mismatched in the miRNA duplex | Fail |
| 23/24 nt miRNA | The miRNA/miRNA* duplex is 23 or 24 nucleotides in length | Warning |
| Asymmetric bulge greater than 3 | There is an asymmetric bulge greater than 3 base pairs in the miRNA duplex | Fail |
| Hairpin is less than 50 nucleotides | The user-provided hairpin sequence is less than 50 nucleotides in length | Fail |
| miRNA multimaps to hairpin | The miRNA or miRNA* provided by the user indexed to multiple locations on the hairpin | Fail |
| Less than 10 reads in a single library | Less than 10 combined miRNA/miRNA* reads in a single library were detected. Does not meet the read floor. | Fail |
| No mature or star reads detected | No reads were detected for the miRNA or miRNA* in a single library | Fail |
| Precision less than 75% | The precision (miRNA reads + miRNA* reads/total reads mapped to hairpin) did not reach 75% in a single library | Fail |
| No 2nt 3' overhang | The user-provided miRNA/miRNA* sequences did not form a duplex with a 2nt 3' overhang | Fail |
| Hairpin structure invalid | The hairpin secondary structure did not allow indexing of miRNA duplex. This may be due to large bulge or secondary stem loop. | Fail |
| Mature miRNA length not met | The mature miRNA length is less than or greater than allowed by criteria | Fail |
| Star length not met | The miRNA star sequence is less than or greater than allowed by criteria | Fail |
| Precursor > 300 nt | The hairpin sequence is larger than 300 nucleotides | Warning (plants) |
| Precursor > 200 nt | The hairpin sequence is larger than 200 nucleotides | Warning (animals) |

## Expression evaluation

The next step of the process is to evaluate expression-based criteria (Table 1). In this phase, miRScore quantifies miRNA abundance and calculates precision for each *MIRNA* locus. Reads from each library are mapped to the hairpin using bowtie version 1.3.1 [36]. All perfect alignments are retained if they align to the forward strand of the putative hairpin sequence. When counting miRNA and miRNA* reads, miRScore allows for one-nucleotide positional variance, which is included to account for biological variation in endonuclease processing during miRNA biogenesis [23]. miRScore requires at least ten reads within a single library to align to the miRNA/miRNA* duplex, allowing for one-nucleotide variants. Raw read counts are used as opposed to normalized values because we are primarily concerned with reproducibility (*i.e.,* observing the miRNA/miRNA* multiple times in a sample) rather than comparative quantification between samples. miRScore then calculates precision for each locus in a library. Precision is defined as the number of reads that map to the miRNA/miRNA* duplex (including one-nucleotide positional variants) divided by the total number of reads which map to the hairpin precursor. The precision threshold is >= 75% (S1 Fig). Only libraries which meet these requirements will have their read count and precision values reported in the results file, but metrics for all libraries can be found within the 'reads.csv' file.

## Identifying potential alternative mature miRNAs in failed loci

Optionally, *MIRNA* loci that fail the initial miRScore analysis are reanalyzed to determine if a different potential mature miRNA exists on the hairpin. This optional procedure is triggered if the user specifies the "-rescue" option in the run command. This feature may be helpful in cases where the initially annotated location of the mature miRNA within the hairpin does not agree with the observed sRNA-seq data. Reanalysis begins by determining the most abundant 20–24 nucleotide sequence that maps to the failed hairpin. miRScore then evaluates this sequence as an 'alternative miRNA' using

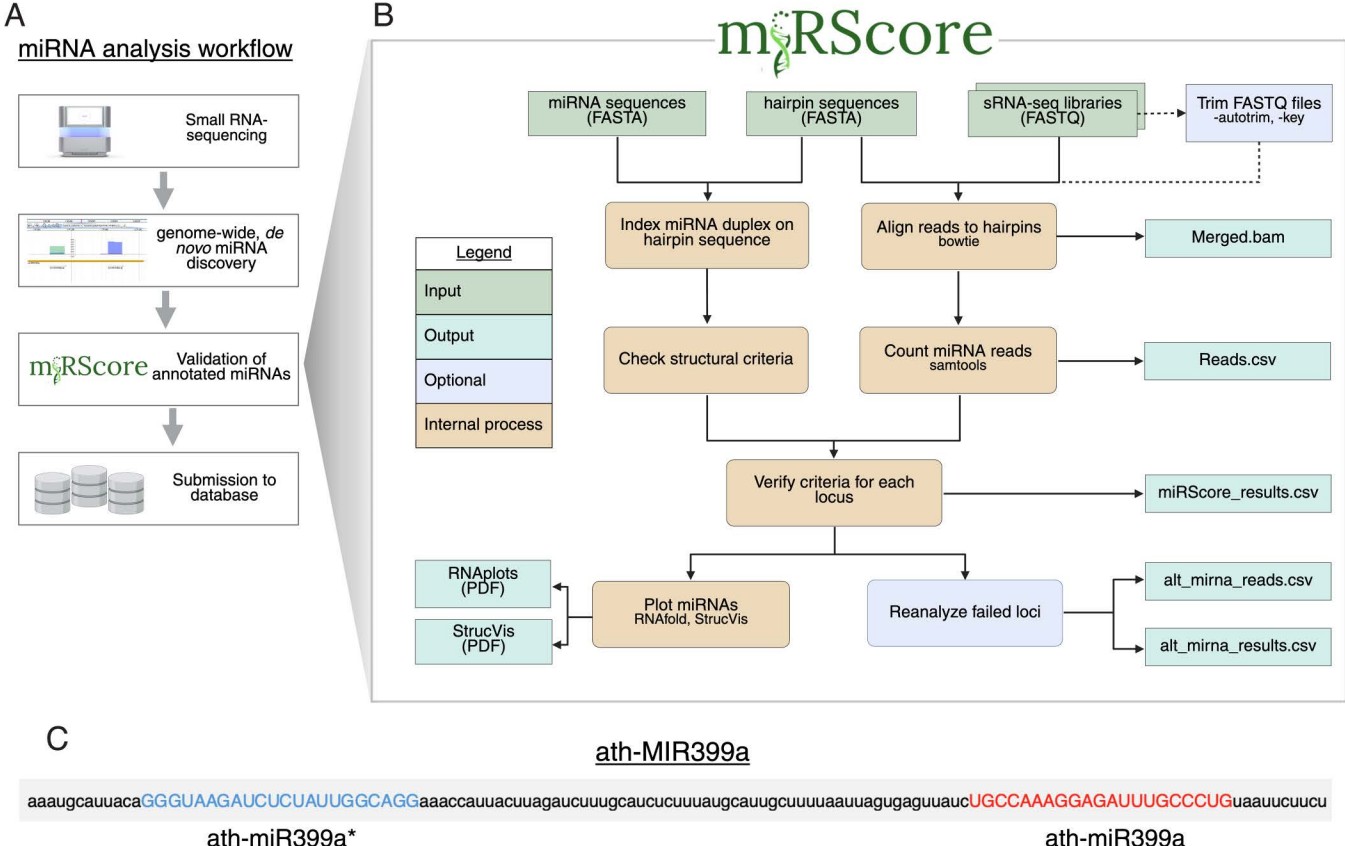

**Fig 1. Workflow and input of miRScore.** (A) miRScore is designed to follow *MIRNA* annotation in the miRNA analysis workflow. (B) Flow chart describing the inputs and steps of miRNA analysis by miRScore (C) Example of suitable names for sequences for input FASTA files. *MIRNA* hairpin identifier (ath-MIR399a) must match the mature miRNA sequence identifier (ath-miR399a); however, the miRNA* (ath-miR399a*) must have an identifier that distinguishes it from the mature miRNA sequence within the file. Created in BioRender [40].

structural and expression criteria (Table 1). If the locus now passes (Table 2), miRScore includes this potential 'alternative miRNA' in a separate alternative results CSV file. Read counts for all alternative miRNAs are reported in an additional alternative reads CSV file. Any potential "rescued" loci that emerge from this optional pipeline should be scrutinized manually before final annotation and submission to a miRNA registry.

## Output and visualization

After assessing structural and expression-based criteria, miRScore generates a CSV file containing details about each locus along with the relevant flags (Table 2) and a pass/fail result. Lastly, miRScore generates figures for each submitted *MIRNA* locus for visualization of secondary structure and read depth.

## Results and discussion

### Performance analysis for annotated MIRNAs

miRScore is primarily designed as a quick secondary filter to analyze new *MIRNA* annotations prior to submission or acceptance into a permanent repository. Because it assesses the validity of an annotation with respect to specific sRNA-seq datasets, it is not appropriate to conclude that a miRNA whose annotation is not supported by specific

datasets is not a *bona fide* miRNA. However, existing repositories contain multitudes of diverse annotations that have already been peer-reviewed and curated and as such are a good source of input data to evaluate the use and performance of miRScore. To this end, we obtained *MIRNA* annotations from miRBase version 22.1 and MirGeneDB version 3.0 from two animal species (*Homo sapiens* and *Mus musculus*) and three plant species from miRBase version 22.1 (*Arabidopsis thaliana, Oryza sativa,* and *Zea mays)* (Table 3). For each plant, five sRNA-seq libraries (S1 File) were acquired from the most frequently cited publication on miRBase that included suitable sRNA-sequencing data [41,42]. sRNA-seq data for animal species were acquired from MirGeneDB website. SRA accession numbers of sRNA-seq data from each species can be found in S1 File. miRScore version 0.3.2 was run using default settings for each dataset.

The primary output of miRScore is a pass/fail result for each locus, accompanied by flags which indicate specific criteria that a *MIRNA* locus did not meet with respect to the provided sRNA-seq datasets (Table 2). A single *MIRNA* locus may receive multiple flags if it fails to meet multiple criteria, and some flags are warnings instead of failures (Table 2). We evaluated the distribution of failed *MIRNA*s across structural and expression-based categories for all tested species (Fig 2A and 2B).

In miRBase animal datasets, many of the failed *MIRNA* loci failed due to both structural and expression criteria (Fig 2A). For example, 885 out of 1615 submitted *H. sapiens MIRNAs* failed due to structural reasons. 526 of these were due to having no 2 nt 3' overhang within the duplex, often off by a single nucleotide (S2 Fig and S2 File). This was observed in several of the failed *MIRNAs* in the MirGeneDB dataset as well (S2 Fig and S2 File). The 2-nt overhang can be more challenging to interpret for miRNA/miRNA* duplexes which contain asymmetric bulges or large sets of

**Table 3. *MIRNA* and small RNA-seq data sources.**

| Organism | *MIRNA* source | Number of sRNA-seq libraries | Citation |
|---|---|---|---|
| *Arabidopsis thaliana* | miRBase v22.1 | 5 | [42] |
| *Oryza sativa* | miRBase v22.1 | 5 | [7] |
| *Zea mays* | miRBase v22.1 | 5 | [41] |
| *Homo sapiens* | miRBase v22.1, MirGeneDB V3.0 | 29 | [31] |
| *Mus musculus* | miRBase v22.1, MirGeneDB V3.0 | 15 | [31] |

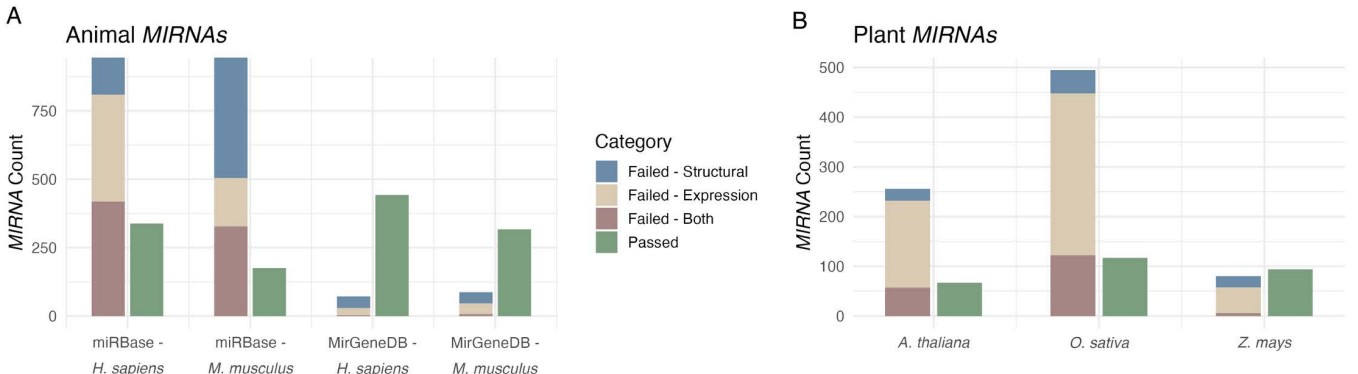

**Fig 2. Performance of miRScore in five annotated *MIRNA* datasets.** (A) miRScore results for animal *MIRNA* datasets from two databases. (Tan) Number of *MIRNAs* which failed miRScore due to expression criteria. (Blue) Number of *MIRNAs* which failed miRScore due to structural criteria. (Mauve) Number of *MIRNAs* which failed miRScore due to both expression and structural criteria. (Green) Number of *MIRNAs* which met all criteria and passed. (B) miRScore results for plant *MIRNA* datasets sourced from miRBase.

mismatches near the ends. For example, hsa-MIR-9-P1 in MirGeneDB has an asymmetric bulge within the annotated 3' overhang (S3 Fig and S2 File). These overhangs are interpreted by miRScore as a 3-nt overhang based on the pairing of the first nucleotide of the duplex on the 5' arm and therefore fails. Most *MIRNAs* from MirGeneDB, which is a curated database, met all criteria within both tested datasets (Figs 2A and S2 and S2 File). For plant *MIRNAs* within miRBase, many failed to meet expression criteria in the given sRNA-seq libraries (Figs 2B and S4 and S3 File). For example, of the 256 failed *A. thaliana MIRNAs*, 173 had no mature or star reads in the analyzed sRNA-seq data, and 41 had a precision of less than 75% (S4 Fig and S3 File). Some of the failures could be attributed to possible tissue-specific or conditional accumulation of the mature miRNA such that the miRNA and/or the miRNA* were absent in the sRNA-seq data used for analysis [24]. These *MIRNAs* meet all structural criteria and would potentially pass given a set of libraries which support expression. Structure failures and some lowly expressed miRNAs could reflect the subset of miRBase annotations that are not true *MIRNAs* [27] or miRNAs made by non-canonical pathways, such as isomiRs and miRtrons.

miRScore visualizations of hairpin secondary structure and read depth for each input locus (Fig 3A–D) allow easy inspection of results with respect to the *MIRNA criteria* (Fig 3E). For example, inspection of the visualizations of *ath-MIR399a* (Fig 3A and 3B), an endogenous *A. thaliana MIRNA*, visually confirms that this locus meets all criteria (Fig 3E). Conversely, *ath-MIR405a* (Fig 3C and 3D) failed miRScore analysis due to unmet expression criteria (Fig 3E).

## Manual validation of annotated MIRNAs

To evaluate miRScore's classification performance, each *MIRNA* locus across all five species was manually inspected to determine its actual condition with respect to the input sRNA-seq datasets (pass or fail). Manual inspection used a combination of data including plots of RNA secondary structure overlaid with annotation and alignment data (Fig 3A–D) and genome browser visualizations of aligned small RNA-seq data. Manual inspection using the criteria defined in Table 1 yielded no observations of false positives or false negatives in any of the analyzed results (Table 4). The difference in the number of true positive and true negative *MIRNAs* in each dataset is striking. There are several factors that affect the number of failed miRNAs. Between 26–58% of miRNAs in the various datasets failed due to expression criteria. In some cases, this is likely due to the small subset of sRNA-seq libraries used to evaluate performance, as miRNA expression can be tissue and condition specific [24,43]. Therefore, when using a more comprehensive set of libraries, these loci may well pass miRScore evaluation. For this reason, it is recommended that miRNAs be validated through miRScore using the same libraries used to annotate them whenever possible. As a corollary, it must be emphasized that these types of "failures" due to lack of accumulation in selected sRNA-seq libraries do not necessarily reflect incorrect annotations in the databases. Painstaking manual curation activities, including literature analysis, will still be required to confirm the validity of many existing annotations. Secondly, the number of miRNAs that fail due to structural criteria in miRBase is likely due to slightly offset annotations of the miRNA/miRNA* duplex positions. One example is hsa-MIR-202, whose miRBase annotation failed, but has an annotation in MirGeneDB which passes. Another reason for the number of structural failures in the animal datasets is the number of loci with mismatches or bulges near the terminal regions of the duplex. This can be challenging and interpretation of the 2-nt 3' overhang may vary in these contexts. For example, hsa-let-7a, which contains an asymmetric bulge at the 5' end of the miRNA, has a miRNA* that is annotated to have what miRScore interprets as a 1-nt 3' overhang (S3 Fig).

## Performance analysis for de novo MIRNAs

miRScore was primarily designed to evaluate new *MIRNA* annotations as a quick screen before or upon submission to databases. One common source of new annotations is those produced by tools that perform genome-wide *de novo* annotation such as ShortStack [17], miRador [19], and miRDeep-P2 [21]. We generated *de novo MIRNA annotation*s in four plant species: *Oryza sativa, Zea mays, Arabidopsis thaliana,* and the parasitic plant *Striga hermonthica. Striga*

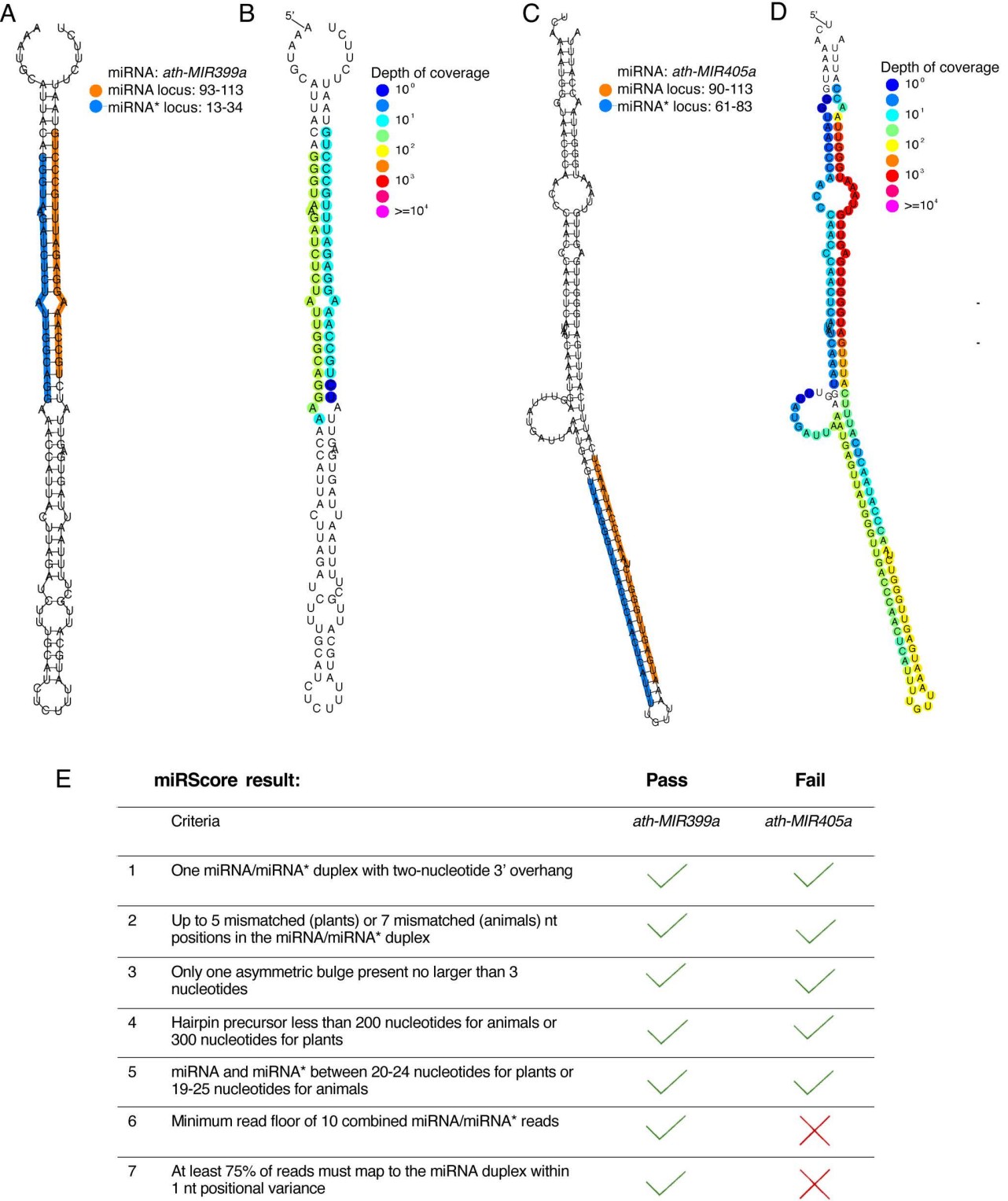

**Fig 3. Visualization of RNA secondary structure and read depth for example *MIRNA*s.** (A) *ath-MIR399a* RNAplot depicting secondary structure. Mature miRNA (orange) and miRNA* (blue) in RNAplot indicate where the user-provided sequence can be found within the hairpin precursor secondary structure. (B) *ath-MIR399a* Strucvis plot depicting read depth of all submitted libraries. (C) *ath-MIR405a* RNAplot depicting secondary structure. (D) *ath-MIR405a* Strucvis plot depicting read depth of all submitted libraries. (E) miRScore criteria and whether each locus met or failed those criteria.

**Table 4. miRScore performance metrics.**

| Organism | *MIRNA*s submitted to miRScore | True positive *MIRNA*s | True negative *MIRNA*s | False positive *MIRNA*s | False negative *MIRNA*s |
|---|---|---|---|---|---|
| *Arabidopsis thaliana* miRBase | 323 | 67 | 256 | 0 | 0 |
| *Oryza sativa* miRBase | 612 | 117 | 495 | 0 | 0 |
| *Zea mays* miRBase | 174 | 94 | 80 | 0 | 0 |
| *Homo sapiens* miRBase | 1615 | 341 | 1274 | 0 | 0 |
| *Mus musculus* miRBase | 1220 | 174 | 1046 | 0 | 0 |
| *Homo sapiens* MirGeneDB | 514 | 442 | 72 | 0 | 0 |
| *Mus musculus* MirGeneDB | 404 | 317 | 87 | 0 | 0 |

*hermonthica* was included as it is currently an unannotated species with no *MIRNA*s cataloged in miRBase or any other database, allowing us to test novel *MIRNA* validation. Each annotation tool was run using the same five libraries used for the plant species in the annotated *MIRNA* dataset (Table 3). For *S. hermonthica*, novel small RNA-seq libraries were generated from leaf and haustorial tissue. miRScore was then run using annotated miRNAs from these results and the sRNA-seq data used for annotation.

miRDeep-P2 annotated the largest number of *MIRNA*s in each species, with over 2,000 *MIRNA*s from *O. sativa* (Fig 4A). Many loci annotated by miRDeep-P2 failed miRScore evaluation (Fig 4B and S4 File). Interestingly, many failed miRNAs had no miRNA* reads, which is a stated requirement for miRDeep-P2 [21]. Nearly all *MIRNA*s annotated by ShortStack passed miRScore inspection (Fig 4B and S4 File). miRador annotations had a pass rate between 80 and 95% with failing loci flagged for various criteria, most of which were expression-based (S4 File). For novel *Striga hermonthica* annotations, both ShortStack and miRador reported 68 passing *MIRNA*s, while miRDeep-P2 reported 127 passing loci.

Unlike annotation software which uses a merged sRNA-seq alignment file to evaluate expression of miRNAs, miRScore evaluates sRNA-seq libraries on an individual basis. This helps account for tissue specificity where a merged library may dilute precision to the point of failure. It also provides a readout of which libraries specifically a miRNA passes in and evidence of replication, which can be useful for downstream analysis. However, this may explain why some miRNAs pass

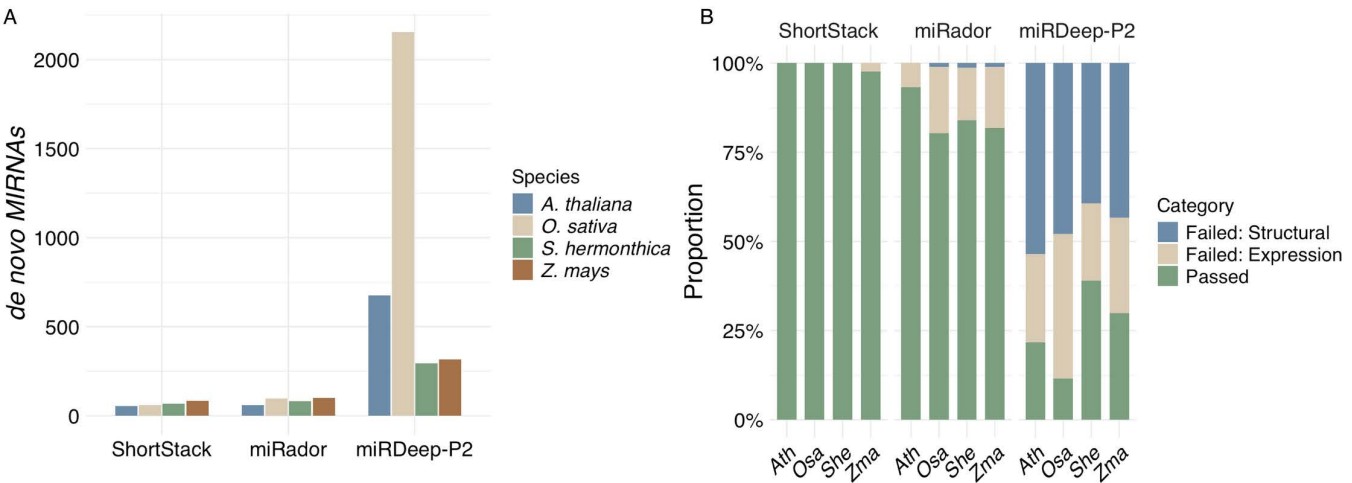

**Fig 4. Evaluation of *de novo MIRNA* annotations in plants.** (A) The number of *MIRNA*s predicted by three *de novo* annotation software by species. (B) Proportion of *de novo MIRNA*s from each software which passed or failed miRScore evaluation in each species including *Arabidopsis thaliana* (Ath), *Oryza sativa* (Osa), *Striga hermonthica* (She), and *Zea mays* (Zma).

during annotation but fail to meet miRScore criteria. It is worth noting that ShortStack, miRador, and miRScore all use RNAfold [35] to predict secondary structure. In addition, the congruence between Shortstack and miRScore at least partially reflects the shared authorship of the two tools. A workflow for each software is included in the supplementary material (S5 File). Overall, miRScore effectively evaluated the outputs of several *MIRNA annotation* tools in plants, confirming its utility in diverse annotation workflows.

## Availability and future directions

miRScore is an open-source python code, and instructions are available on GitHub at https://github.com/Aez35/miRScore. A Bioconda recipe for miRScore is available, allowing easy installation using the conda package manager [37]. In this study, we demonstrate that miRScore effectively validates *MIRNAs* in both annotated and novel *MIRNA* datasets across plant and animal species. miRScore enables rapid and robust analysis of *MIRNA* loci, without requiring a reference genome, and offers detailed metrics and visualizations for each locus to support comprehensive analysis of *MIRNA* datasets. We believe that miRScore will contribute to improving *MIRNA* annotation quality and provide a tool for researchers to quickly verify annotations prior to downstream analysis. In addition, miRScore will provide an automated validation tool which will reduce the time it takes to validate novel *MIRNAs* prior to publication and database submission. miRScore's ability to identify high confidence *MIRNAs* using widely accepted criteria quickly and accurately provides a valuable tool, contributing to the enhancement of *MIRNA* annotation quality in future studies.

## Methods

### Annotated datasets

Annotated *MIRNAs* from *Oryza sativa, Arabidopsis thaliana, Zea mays, Mus musculus,* and *Homo sapiens* were tested using miRScore version 0.3.2. Mature miRNA and hairpins sequences for each species were downloaded from either miRBase version 22.1 or MirGeneDB version 3.0. The miRNA file submitted to miRScore for each species contained sequences for both the annotated miRNA and miRNA* sequence where applicable. Several miRNAs from miRBase in some species began at position one of their respective hairpins and did not allow for evaluation of the miRNA duplex, particularly the two-nucleotide 3' overhang structure. In these instances, miRScore will report the offending sequences and quit the run. These miRNAs were therefore removed from the dataset prior to running miRScore. This was not an issue with miRNAs sourced from MirGeneDB, as the database included an option to download extended precursors. Both miRBase and MirGeneDB data require some processing to harmonize naming conventions prior to miRScore analysis: Details are provided in S6 File. sRNA-seq data accession numbers and sources used for evaluation are listed in Table 3 and S1 File.

### Processing and alignment of small RNA-seq data

Small RNA-seq data for each dataset were trimmed to remove 3' adapters using the '-autotrim' feature of miRScore. Trim keys for each dataset were set to the miRScore default of ath-miR166a (UCGGACCAGGCUUCAUUCCCC) for plants and hsa-let-7a (UGAGGUAGUAGGUUGUAUAGUU) for animals. The trimmed sRNA-seq data were aligned to hairpin sequences using bowtie [36] version 1.3.1 with options '-v0 -a –no-unal –norc'. The BAM alignment files were merged and read groups were used to count reads that mapped to each hairpin using samtools [34] version 1.20.

### Striga hermonthica growth and sRNA-seq library preparation

*Striga hermonthica* Kibos ecotype was grown on host *Oryza sativa* ssp. japonica variety Kitaake under 16-hour light conditions in a quarantine facility at 30°C for 45 days. Haustorium and leaf tissue were collected from *S. hermonthica* and total

RNA was extracted using Zymo Quick-RNA Plant Miniprep Kit. sRNA-seq libraries were prepared essentially as described in Maguire et al. [44]. Sequencing of the prepared libraries was performed on an Illumina NextSeq2000. New small RNA-seq libraries from *S. hermonthica* have been deposited at NCBI GEO under accession GSE282265.

**de novo MIRNA datasets**

*MIRNAs* were annotated in four plant species using ShortStack version 4.0.4 [17], miRador commit c68c153 [19], and miRDeep-P2 version 1.1.4 [21] (See S5 File). All annotation software were run on default settings for *de novo MIRNA* discovery. Genome assembly versions were: *Arabidopsis thaliana* (TAIR 10), *Oryza sativa* (IRGSP-1.0)*, Striga hermonthica* assembly SHERM (GCA_902706635.1)*, and Zea mays* (Zm-B73-REFERENCE-NAM-5.0). sRNA-seq data for *Striga hermonthica* was generated as described above. sRNA-seq data for the other plant species were acquired from accession numbers in S1 File. Mature miRNA and hairpin sequences from resulting annotations were parsed and saved to two separate FASTA files. These FASTA files were used to test validation of *de novo MIRNAs* using miRScore version 0.2.0. The same sRNA-seq data used for *MIRNA annotation* were used to run miRScore for each species.

**Plots**

Data plotted in Figs 2 and 4 are taken from S1, S2, and S3 Files. Code used to produce the actuals plots is given in S7 File.

**Supporting information**

**S1 File. SRA accession numbers of sRNA-seq libraries used for testing miRScore on each dataset.**
(XLSX)

**S2 File. miRScore results file for miRNA datasets from MirGeneDB and miRBase for two animal species (*Homo sapiens* and *Mus musculus*).**
(XLSX)

**S3 File. miRScore results file for miRNA datasets from MirGeneDB and miRBase for three plant species (*Arabidopsis thaliana*, *Oryza sativa*, and *Zea mays*).**
(XLSX)

**S4 File. miRScore results files from *de novo MIRNAs* from three annotation software of four plant species.**
(XLSX)

**S5 File. Markdown pdf file describing *de novo* annotation pipeline for plant miRNAs used to test miRScore handling of *de novo* annotations.**
(PDF)

**S6 File. Markdown file describing how to prepare miRBase and MirGeneDB data for running miRScore.**
(PDF)

**S7 File. R script file for generating Figs 2 and 4.** Format: Plain text/ R code (.R).
(R)

**S1 Fig. Explanation of read alignment, precision, and variance.** (A) When counting miRNA duplex reads, a variance window of -/+1 nt from the indexed start/stop position of the miRNA and miRNA*. Reads which start and stop within this window are counted towards the total miRNA duplex count and used to determine precision. (B) Example of reads which are included in total count of hsa-mir-212 miRNA (red), miRNA* (blue), and those that are not included in count (black).

Read length (len) and number of reads aligned at that position (al) can be found on the right side. (C) Example of reads included in ath-MIR167a miRNA (red), miRNA* (blue), and those that are not included (black).
(DOCX)

**S2 Fig. Upset plot of result and flags for *MIRNAs* sourced from miRBase and MirGeneDB for animal species.** See S2 File for source data. (A) *Mus musculus* (mmu) mirbase *MIRNA* results and flags. (B) *Homo sapiens* (hsa) mirbase *MIRNA* results and flags. (C) mmu MirGeneDB *MIRNA* results and flags. (D) hsa MirGeneDB *MIRNA* results and flags.
(DOCX)

**S3 Fig. RNAplots of *MIRNA* secondary structures.** (A) Plot of Hsa-Mir-9-P1 from MirGeneDB with annotated miRNA (orange) and miRNA* (blue). (B) Plot of hsa-let-7a-1 from miRBase with annotated miRNA (orange) and miRNA* (blue).
(DOCX)

**S4 Fig. Upset plot of results and flags for *MIRNAs* sourced from miRBase for plant species.** See S3 File (A) *Arabidopsis thaliana* (ath) *MIRNA* results and flags. (B) *Oryza sativa* (osa) *MIRNA* results and flags. (C) *Zea mays* (zma) *MIRNA* results and flags.
(DOCX)

## Acknowledgments

We thank the Penn State Genomics Core Facility (RRID: SCR_023645) for sRNA-seq services. We thank Steven Runo and Claude dePamphilis for the gift of *Striga hermonthica* seed.

## Author contributions

**Conceptualization:** Allison Vanek, Sam Griffiths-Jones, Blake C. Meyers, Michael J. Axtell.

**Data curation:** Allison Vanek, Michael J. Axtell.

**Formal analysis:** Allison Vanek.

**Funding acquisition:** Sam Griffiths-Jones, Blake C. Meyers, Saima Shahid, Michael J. Axtell.

**Methodology:** Allison Vanek, Michael J. Axtell.

**Project administration:** Sam Griffiths-Jones, Blake C. Meyers, Michael J. Axtell.

**Software:** Allison Vanek, Michael J. Axtell.

**Supervision:** Michael J. Axtell.

**Visualization:** Allison Vanek.

**Writing – original draft:** Allison Vanek, Michael J. Axtell.

**Writing – review & editing:** Allison Vanek, Michael J. Axtell.

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
