## [Decision Letter · Decision Letter 0]

2 Apr 2025

PCOMPBIOL-D-24-02180

miRScore: a rapid and precise microRNA validation tool

PLOS Computational Biology

Dear Dr. Axtell,

Thank you for submitting your manuscript to PLOS Computational Biology.The manuscript was overall well received by the reviewers and considered to be of wider interest, however they each noted weaknesses in the work and areas in need of clarification. In some parts, a better recognition of the existing literature is required. Reviewers noted that if the results strongly disagree with reference databases, then a higher level of evidence is required. Moreover the rigour of the work could be improved by adopting enhanced computational reproducibility practices. A more detailed comparison to existing tools/databases and thorough discussion of the discrepancies is required. 

We invite you to submit a revised version of the manuscript that addresses the points raised during the review process.

Please submit your revised manuscript within 60 days Jun 02 2025 11:59PM. If you will need more time than this to complete your revisions, please reply to this message or contact the journal office at ploscompbiol@plos.org. Please include the following items when submitting your revised manuscript:

We look forward to receiving your revised manuscript.

Kind regards,

Francisco Zorrilla

Guest Editor

PLOS Computational Biology

Shihua Zhang

Section Editor

PLOS Computational Biology

**Journal Requirements:**

5) Please amend your detailed Financial Disclosure statement. This is published with the article. It must therefore be completed in full sentences and contain the exact wording you wish to be published. State what role the funders took in the study. If the funders had no role in your study, please state: "The funders had no role in study design, data collection and analysis, decision to publish, or preparation of the manuscript.".

**Reviewers' comments:**

Reviewer's Responses to Questions

**Comments to the Authors:**

Reviewer #1: Vanek et al (miRScore: a rapid and precise microRNA validation tool) present a computational tool mirScore for assessing the validity of microRNA predictions in plants and animals. Based on 5 selected species they show that the vast majority of annotated microRNAs in miRBase are incorrect (Figure 2) and also test novel candidates in 4 species (Fig 4).

While the tool seems useful, it is unclear if the authors imply that human has only 92 microRNAs (Table3)?

Regardless, the lack of scholarship in ignoring the literature and efforts by others such as Bracken (e.g. https://www.biorxiv.org/content/10.1101/2024.12.05.626958v1.abstract), Corey (e.g. https://www.biorxiv.org/content/10.1101/2022.10.18.512653v1.abstract) and our work (discloser: I am Bastian Fromm), especially regarding MirGeneDB & annotation criteria for animals is sobering (e.g. https://rnajournal.cshlp.org/content/28/6/781.short & https://www.annualreviews.org/content/journals/10.1146/annurev-genet-120213-092023). See also refs below

Given that the authors SGJ and MJA run their own databases, it is further not understandable why the direct consequences of the proposed tool: the high false-positives in miRBase and possibly the other database, are not highlighted and acted upon by cleaning them up.

In order to acomodate my critiques, the authors could run their tool on MirGeneDB species. In our latest release (Clarke 2025, NAR), we provide smallRNA seq data.

1. Castellano, L and Stebbing, J (2013). Deep sequencing of small RNAs identifies canonical and non-canonical miRNA and endogenous siRNAs in mammalian somatic tissues. Nucleic Acids Res. 41: 3339–3351.

2. Chiang, HR, Schoenfeld, LW, Ruby, JG, Auyeung, VC, Spies, N, Baek, D, et al. (2010). Mammalian microRNAs: experimental evaluation of novel and previously annotated genes. Genes Dev. 24: 992–1009.

3. Jones-Rhoades, MW (2012). Conservation and divergence in plant microRNAs. Plant Mol. Biol. 80: 3–16.

4. Ludwig, N, Becker, M, Schumann, T, Speer, T, Fehlmann, T, Keller, A, et al. (2017). Bias in recent miRBase annotations potentially associated with RNA quality issues. Sci. Rep. 7: 5162.

5. Langenberger, D, Bartschat, S, Hertel, J, Hoffmann, S, Tafer, H and Stadler, PF (2011). MicroRNA or Not MicroRNA? Advances in Bioinformatics and Computational Biology, Springer Berlin Heidelberg: pp 1–9.

6. Meng, Y, Shao, C, Wang, H and Chen, M (2012). Are all the miRBase-registered microRNAs true? A structure- and expression-based re-examination in plants. RNA Biol. 9: 249–253.

7. Tarver, JE, Donoghue, PC and Peterson, KJ (2012). Do miRNAs have a deep evolutionary history? Bioessays 34: 857–866.

8. Taylor, RS, Tarver, JE, Hiscock, SJ and Donoghue, PC (2014). Evolutionary history of plant microRNAs. Trends Plant Sci. doi:10.1016/j.tplants.2013.11.008.

9. Wang, X and Liu, XS (2011). Systematic Curation of miRBase Annotation Using Integrated Small RNA High-Throughput Sequencing Data for C. elegans and Drosophila. Front. Genet. 2: 25.

10. Fromm, B, Billipp, T, Peck, LE and Johansen, M (2015). A uniform system for the annotation of vertebrate microRNA genes and the evolution of the human microRNAome. Annual review ofat <http: 10.1146="" abs="" annurev-genet-120213-092023="" doi="" www.annualreviews.org="">.

11. Guo, Z, Kuang, Z, Wang, Y, Zhao, Y, Tao, Y, Cheng, C, et al. (2020). PmiREN: a comprehensive encyclopedia of plant miRNAs. Nucleic Acids Res. 48: D1114–D1121.

12. Fromm, B, Domanska, D, Høye, E, Ovchinnikov, V, Kang, W, Aparicio-Puerta, E, et al. (2019). MirGeneDB 2.0: the metazoan microRNA complement. Nucleic Acids Res.: 258749.

13. Fromm, B, Keller, A, Yang, X, Friedlander, MR, Peterson, KJ and Griffiths-Jones, S (2020). Quo vadis microRNAs? Trends Genet. 36: 461–463.

14. Fromm, B, Høye, E, Domanska, D, Zhong, X, Aparicio-Puerta, E, Ovchinnikov, V, et al. (2022). MirGeneDB 2.1: toward a complete sampling of all major animal phyla. Nucleic</http:>

Reviewer #2: # Review for Vanek et al 2025

This manuscript describes miRScore, a software allowing quality control of novel or existing miRNA annotations. In that extent, it uses both structural features and sRNA-seq libraries to assess the quality of miRNA annotations. It is demonstrated by applying it on both existing annotations from the miRBase database and novel annotations produced by state of the art software. It is a welcome addition to the miRNA sofware tooltip, given the increasing number of false positive miRNAs submitted to databases. The paper is well written and does a good job of explaining how the software works However, I have two main concerns that I'd like to be at least partially adressed before considering the paper ready for publication

## Main comments :

### Comparison to mirBase annotations

The first part of the results demonstrate the pertinence of the software when applied on the miRBase database, which is the generalist reference database for microRNAs.

On this database, for four model species, miRScore dives a "Pass" to between 6 and 54% of registered sequences in miRBase. Annotations failing because of low expression are easily understandable because of the low number of samples (5) used to validate them.

However, the data shows that more than 50% of annotated miRNAs in miRBase for mouse and human don't have a correct miRNA structure (mostly because of missing 3' overhang). Since miRBase is regarded as a reference database, I think the author need to better support these results, and I have the following remarks :

- Looking at a failing MIRNA like hsa-mir-25, and at the code, I understand that you require a 2nt overhang to have a matching sequence on the hanging part. I thought it was not necessary to see an overhang here but I'm not sure.

- While mirBase is very used, they are actually other databases that undewent more curation. In mirGeneDB, I could find at least two genes (hsa-mir-202 and hsa-mir-511) whose mature/star where corrected and show an overhang. I think it would make a lot of sense to run miRScore mirGeneDB, with an opportunity to use miRScore to highlight the difference between miRBase and a curated database. I'm also very pleased to see that hsa-mir-202 has an alternative version in Supp S2 that matches mirGeneDB, nice. Maybe it's something to follow to demonstrate how miRScore can improve an annotation?

## Reproducibility of the results

While the GitHub repository includes a working minimal example, I think the paper would benefit of making the results of the paper more reproducible. I tried on my side to run miRScore on the full miRNA repertoire of a species in miRBase, and could not easily do it (see log below), maybe because I missed some preprocessing step? Could you please include in the documentation a complete example of how to run miRscore on a mirBase species just as in the paper?

```

---

miRScore version 0.3.2

Options:

'Mature file' ath.mature.fa

'Hairpin file' ath.hairpin.fa

'Threads' 1

'Kingdom' plant

'Fastqs' ['fastq/SRR218085.fastq.gz', 'fastq/SRR218092.fastq.gz', 'fastq/SRR218096.fastq.gz', 'fastq/SRR218098.fastq.gz', 'fastq/SRR218099.fastq.gz']

'Output directory' miRScore_output/

---

miRNAs submitted: 326

Checking hairpin and miRNA sequences...

Traceback (most recent call last):

File "/home/cguyomar/conda/env_mirscore/bin/miRScore", line 1557, in <module>

main()

File "/home/cguyomar/conda/env_mirscore/bin/miRScore", line 913, in main

star=hairpin[(starpos[0]-1):starpos[1]].upper()

~~~~~~~~~~^~

TypeError: unsupported operand type(s) for -: 'NoneType' and 'int'

```

Also, for the sake of reproducibility, it would be nice to deposit somewhere :

- The outputs and commands of ShortStack/miRador/mirDeep-P2

- The scripts to generate the figures of the paper

# Minor comments

l168 : Could you please give the options used for Bowtie (if different from default)

l172 : You are using a raw 10 read expression threshold. Shouldn't you perform a minimal amount of normalization like RPM to account for library size differences?

l173 : It is actually common that the passenger strand is not detected at all. While I understand the concern, I find that "No star reads detected" could deserve its own flag, less worrying as "No mature or star reads detected".

l174 : Could precision be impacted by isomirs? Is an exact matching required to consider a read as "miRNA duplex read"?

l236 : I'm not sure to understand why is reference 38 here. I understand it is about the evolutionary dynamics of microRNAs and their targets and not their structural features? I think what could be mentionned here is non canonical miRNAs pathway like mirtrons. Could they be a part of the explanation for the many missing 3' overhangs?

l275 : Since ShortStack has been developped by the last author, could I be that it already includes structural constraints analog to those of miRScore when predicting MIRNAs?</module>

Reviewer #3: 1) The theme of the manuscript 'miRScore: a rapid and precise microRNA validation tool' is relevant to the Journal.

2) The basis of Criteria for endogenous miRNAs in plants and animals as mentioned in Table 1 needs clarification. Demonstration with specific known miRNA in different species can be conducted to illustrate stringency of the criteria. Though similar attempt with group of miRNA has been conducted as mentioned in Table 3.

3) Performance of validation of miRDeep-2 versus Shortstack merits discussion.

4) A comparative table of performance of miRScore with entries from difference databases will be useful.

**Have the authors made all data and (if applicable) computational code underlying the findings in their manuscript fully available?**

Reviewer #1: Yes

Reviewer #2: **No: ** The code to reproduce the paper results is missing

Reviewer #3: Yes

PLOS authors have the option to publish the peer review history of their article (what does this mean? ). If published, this will include your full peer review and any attached files.

**Do you want your identity to be public for this peer review?** For information about this choice, including consent withdrawal, please see our Privacy Policy .

Reviewer #1: **Yes: ** Bastian Fromm

Reviewer #2: No

Reviewer #3: **Yes: ** Ashwin Kotnis

**Figure resubmission:**

**Reproducibility:**



---

## [Decision Letter · Decision Letter 1]

15 Jul 2025

miRScore: a rapid and precise microRNA validation tool

PLOS Computational Biology

Dear Dr. Axtell,

Thank you for submitting your manuscript to PLOS Computational Biology. After careful consideration, we feel that it has merit but does not fully meet PLOS Computational Biology's publication criteria as it currently stands. Therefore, we invite you to submit a revised version of the manuscript that addresses the points raised during the review process.

Please submit your revised manuscript within 60 days Sep 14 2025 11:59PM. If you will need more time than this to complete your revisions, please reply to this message or contact the journal office at ploscompbiol@plos.org. Please include the following items when submitting your revised manuscript:

We look forward to receiving your revised manuscript.

Kind regards,

Mark Ziemann

Academic Editor

PLOS Computational Biology

Ilya Ioshikhes

Section Editor

PLOS Computational Biology

**Reviewers' comments:**

Reviewer's Responses to Questions

**Comments to the Authors:**

Reviewer #1: I am happy to see the authors have added two species from MirGeneDB as a comparison, but I am sceptical whether the functionality of the tool "a tool to quickly analyze novel and annotated MIRNAs for further analysis or submission to a database" has sufficiently been shown, or, for this matter, is actually applied by the authors themselves.

The listed annotation criteria (Table 1) seem arbitrary and - for animals - somewhat out of touch with studies by Bartel, Kim and also us on annotation criteria for animals. For example did we show that mature microRNAs in animals can be between 20 and 27 nucleotides in length (Fromm 2015 AnnRev) and that there is no "hard" limit on microRNA hairpin lengths in animals (Fromm 2013, MBE and e.g. see also https://mirgenedb.org/show/dpu/Mir-375. Either way: where does 200nt and 300nt for max length come from? Where the other criteria?

For the plant microRNA part, I am aware that there are at least 3 competing microRNA databases for plants (including miRBase), but it seems the annotations are not compared to other databases, why?

I am curious why are you only looking at so few species, and why at two very closely related species for animals? Why canæt you do all miRBase and plant DB species?

Finally, to come back to the scope of the tool to be informing on existing databases' annotations: the authors (SGJ, BCM, MJA) host leading microRNA / sncRNA repositories that are only partially tested here and they found for miRBase massive numbers of false-positives (as many others did before) - Why are the authors not acting on their results and remove those false entries from their DBs?

The inclusion of false-positives has had a really negative influence on the field and miRScore is the perfect opportunity to rectify this.

MINOR: please spell MirGeneDB consistently MirGeneDB.

Reviewer #2: I am satisfied with the revisions made by the authors, which have significantly improved the quality of the paper. I have no further comments at this time.

**Have the authors made all data and (if applicable) computational code underlying the findings in their manuscript fully available?**

Reviewer #1: Yes

Reviewer #2: Yes

PLOS authors have the option to publish the peer review history of their article (what does this mean? ). If published, this will include your full peer review and any attached files.

**Do you want your identity to be public for this peer review?** For information about this choice, including consent withdrawal, please see our Privacy Policy .

Reviewer #1: **Yes: ** Bastian Fromm

Reviewer #2: No

**Figure resubmission:**
---

## [Decision Letter · Decision Letter 2]

8 Sep 2025

PCOMPBIOL-D-24-02180R2

miRScore: a rapid and precise microRNA validation tool

PLOS Computational Biology

Dear Dr. Axtell,

Thank you for submitting your manuscript to PLOS Computational Biology. After careful consideration, we feel that it has merit but does not fully meet PLOS Computational Biology's publication criteria as it currently stands. Therefore, we invite you to submit a revised version of the manuscript that addresses the points raised during the review process.

Please submit your revised manuscript within 60 days Nov 08 2025 11:59PM. If you will need more time than this to complete your revisions, please reply to this message or contact the journal office at ploscompbiol@plos.org. Please include the following items when submitting your revised manuscript:

We look forward to receiving your revised manuscript.

Kind regards,

Mark Ziemann

Academic Editor

PLOS Computational Biology

Ilya Ioshikhes

Section Editor

PLOS Computational Biology

**Additional Editor Comments :**

As this article has been in the review process for a long time, I am keen to provide straightforward direction to progress it hopefully to acceptance quickly.

As the reviewer states, the rescue feature of miRscore appears to be identifying some problematic loci as true miRNA genes. In fact a large fraction (~50%) of recently rejected miRbase microRNAs appear to be true miRNAs using miRscore. As the reviewer states, this could cause confusion in the field. Therefore for the benefit of the field, miRscore criteria should be adjusted to exclude the known problematic loci including the ones named by the reviewer. As hard thresholds are rarely perfect classifiers, authors could consider tiered classification (eg: high confidence, low confidence and no-miR loci) where the doubtful miRs could be placed in the low confidence group.

Authors could also include a passage in the introduction about the clean-up of miRbase to remove some false/problematic miRs.

**Reviewers' comments:**

Reviewer's Responses to Questions

**Comments to the Authors:**

**Please note that one review is uploaded as an attachment.**

Reviewer #1: I was reluctant to follow the argument about applying miRscore to the whole of miRBase or other plant databases, but could see (from experience) that this would be a lot of work and, hence, was to be OK with the revisions. However, then I noticed the "rescued" miRBase microRNAs and had a brief look at the entries rescued which really was surprising and a bit shocking to me:

Not only were there many rescued "microRNAs" previously rejected by us , showing that miRscore actually could be detrimental to previous efforts to clean up the number of misannotated human loci, but in addition (highlighted in the figure I uploaded) were there clear cases of very bad examples, such as "mir-548" (a transposon with many hits), "mir-1258" with more than hundred hits in the genome, mir-3130 with a palindromic (very unusual!) sequence and mir-6132 with mature reads in mirbase below 20nts....

I think the rescuing feature is counterintuitive and will open floodgates of hundreds of novel "miRNAs" to be validated with miRscore - something that shoulg be avoided for the sake of the field!

**Have the authors made all data and (if applicable) computational code underlying the findings in their manuscript fully available?**

Reviewer #1: Yes

PLOS authors have the option to publish the peer review history of their article (what does this mean? ). If published, this will include your full peer review and any attached files.

**Do you want your identity to be public for this peer review?** For information about this choice, including consent withdrawal, please see our Privacy Policy .

Reviewer #1: **Yes: ** Bastian Fromm

**Figure resubmission:**

**Reproducibility:**



---

## [Decision Letter · Decision Letter 3]

19 Oct 2025

PCOMPBIOL-D-24-02180R3

miRScore: a rapid and precise microRNA validation tool

PLOS Computational Biology

Dear Dr. Axtell,

Thank you for submitting your manuscript to PLOS Computational Biology. After careful consideration, we feel that it has merit but does not fully meet PLOS Computational Biology's publication criteria as it currently stands. Therefore, we invite you to submit a revised version of the manuscript that addresses the points raised during the review process.

Please submit your revised manuscript within 30 days Dec 19 2025 11:59PM. If you will need more time than this to complete your revisions, please reply to this message or contact the journal office at ploscompbiol@plos.org. Please include the following items when submitting your revised manuscript:

We look forward to receiving your revised manuscript.

Kind regards,

Mark Ziemann

Academic Editor

PLOS Computational Biology

Ilya Ioshikhes

Section Editor

PLOS Computational Biology

**Reviewers' comments:**

Reviewer's Responses to Questions

Reviewer #1: I appreciate that the authors took my criticism seriously. We want to improve the standards in the field, not make it worse. To this end the authors should remove Figure 2C and D.

**Have the authors made all data and (if applicable) computational code underlying the findings in their manuscript fully available?**

Reviewer #1: None

PLOS authors have the option to publish the peer review history of their article (what does this mean? ). If published, this will include your full peer review and any attached files.

**Do you want your identity to be public for this peer review?** For information about this choice, including consent withdrawal, please see our Privacy Policy .

Reviewer #1: **Yes: ** Bastian Fromm

**Figure resubmission:**
---

## [Decision Letter · Decision Letter 4]

25 Oct 2025

Dear Prof. Axtell,

We are pleased to inform you that your manuscript 'miRScore: a rapid and precise microRNA validation tool' has been provisionally accepted for publication in PLOS Computational Biology.

Best regards,

Mark Ziemann

Academic Editor

PLOS Computational Biology

Ilya Ioshikhes

Section Editor

PLOS Computational Biology

Reviewer's Responses to Questions

**Comments to the Authors:**

Reviewer #1: well done

**Have the authors made all data and (if applicable) computational code underlying the findings in their manuscript fully available?**

Reviewer #1: None

PLOS authors have the option to publish the peer review history of their article (what does this mean? ). If published, this will include your full peer review and any attached files.

**Do you want your identity to be public for this peer review?** For information about this choice, including consent withdrawal, please see our Privacy Policy .

Reviewer #1: **Yes: ** Bastian Fromm

---

## [Editor Report · Acceptance letter]

PCOMPBIOL-D-24-02180R4

miRScore: a rapid and precise microRNA validation tool

Dear Dr Axtell,

I am pleased to inform you that your manuscript has been formally accepted for publication in PLOS Computational Biology. Your manuscript is now with our production department and you will be notified of the publication date in due course.

With kind regards,

Judit Kozma
